# A machine learning approach to model the impact of line edge roughness on gate-all-around nanowire FETs while reducing the carbon footprint

Antonio García-Loureiro[1][*], Natalia Seoane[1], Julián G. Fernández[1], Enrique Comesaña[2], Juan C. Pichel[1]

**1** CITIUS, Universidade de Santiago de Compostela, Santiago de Compostela, Spain, **2** Departamento de Electrónica e Computación, Universidade de Santiago de Compostela, Lugo, Spain

☯ These authors contributed equally to this work.
* antonio.garcia.loureiro@usc.es

**Data Availability Statement:** The data and source codes that support the findings of this study are

## Abstract

The performance and reliability of semiconductor devices scaled down to the sub-nanometer regime are being seriously affected by process-induced variability. To properly assess the impact of the different sources of fluctuations, such as line edge roughness (LER), statistical analyses involving large samples of device configurations are needed. The computational cost of such studies can be very high if 3D advanced simulation tools (TCAD) that include quantum effects are used. In this work, we present a machine learning approach to model the impact of LER on two gate-all-around nanowire FETs that is able to dramatically decrease the computational effort, thus reducing the carbon footprint of the study, while obtaining great accuracy. Finally, we demonstrate that transfer learning techniques can decrease the computing cost even further, being the carbon footprint of the study just 0.18 g of $CO_2$ (whereas a single device TCAD study can produce up to 2.6 kg of $CO_2$), while obtaining coefficient of determination values larger than 0.985 when using only a 10% of the input samples.

## Introduction

In nanoelectronics, an unsolved issue is the ever-closer limit of transistor scaling that threatens to put a halt to the digital revolution observed over the last 50 years [1]. Therefore, it is essential and urgent to investigate new alternatives and solutions to be used in future transistor technology nodes. Currently, gate-all-around (GAA) device architectures, like nanosheet (NS) or nanowire (NW) FETs, are suggested as strong contenders by the International Roadmap for Devices and Systems [2], because of their excellent electrostatic control [3].

Considering that the fabrication of nanoelectronic devices is a long, complex and very expensive process [4], the use of Technology Computer-Aided Design (TCAD) to predict device performance is mandatory in order to reduce costs and to optimize development times [5]. At the nanoscale, the random deficiencies introduced during the manufacturing process

openly available in open access at http://doi.org/10.5281/zenodo.7674909 in the Zenodo Repository.

**Funding:** Work supported by the Spanish Ministerio de Ciencia e Innovación (grants RYC-2017-23312, PID2019-104834GB-I00, PLEC2021-007662) and by Xunta de Galicia and FEDER Funds (grants, ED431F 2020/008 and ED431C 2022/16). The funders had no role in study design, data collection and analysis, decision to publish, or preparation of the manuscript.

**Competing interests:** The authors have declared that no competing interests exist.

lead to variability issues, heavily impacting the performance and reliability of the final product. Metal-gate granularity (MGG), line edge roughness (LER), random discrete dopants (RDD), oxide thickness variation (OTV) and interface trap charges (ITC) are the main sources of variability affecting current multigate transistors [6]. To properly analyze the effect of these sources of fluctuations, statistical analysis of large ensembles of devices are needed [7]. On top of that, three-dimensional simulations that account for quantum effects are required to realistically model device behavior [8], heavily increasing the computational cost of the studies. For that reason, it is relevant to apply complementary techniques, such as machine learning (ML) [9, 10], to either shorten the computational times or to open the path to the investigation of other effects that would be unfeasible using only TCAD. Recently, different aspects of machine learning have attracted interest in the field of nanoelectronics. At circuit level, ML techniques have been applied to predict the current-voltage curves needed for NW FETs compact models [11]. At device level, several works have analyzed the impact of MGG or/and RDD induced variability in GAA NW FETs [12, 13] and NS FETs [14, 15]. However, other sources of variability, such as LER, have not been investigated so far.

Within this work, we demonstrate that multi-layer perceptron networks can efficiently predict the effect of LER in state-of-the-art GAA NW FETs, greatly reducing the number of device simulations required to fully capture this effect and thus, the associated computational cost. In addition, we evidence that the use of transfer learning techniques can further decrease the computing effort, obtaining coefficient of determination values ($R^2$) above 0.985 when using only a 10% of the input samples.

## Methodology

Fig 1 shows 2D cross-sectional schematics of the two Si-based GAA NW FETs used in this work, a 22 nm gate length device (top figures) and a 10 nm gate length one (bottom figures). Their main device dimensions are included in Table 1 for an easy comparison. These devices have an uniform p-type doping in the semiconductor channel and a n-type Gaussian doping in the source/drain (s/d) regions, that is fixed to $N_{s/d}$ from the s/d contacts till a point ($X_m$) nearby the gate region, where the doping exponentially decays (with a slope $\delta$), as shown in Fig 2. The specific doping values for each device and region are also included in Table 1.

The 22 nm GAA NW FET structure is designed after an experimental device [16] and, Fig 3 shows a comparison of experimental versus simulated $I_D$-$V_G$ characteristics, on both linear and logarithmic scales, at a supply bias of 1.0 V. Two three-dimensional finite-element (FE) device simulation approaches, implemented in VENDES [17], have been considered in this

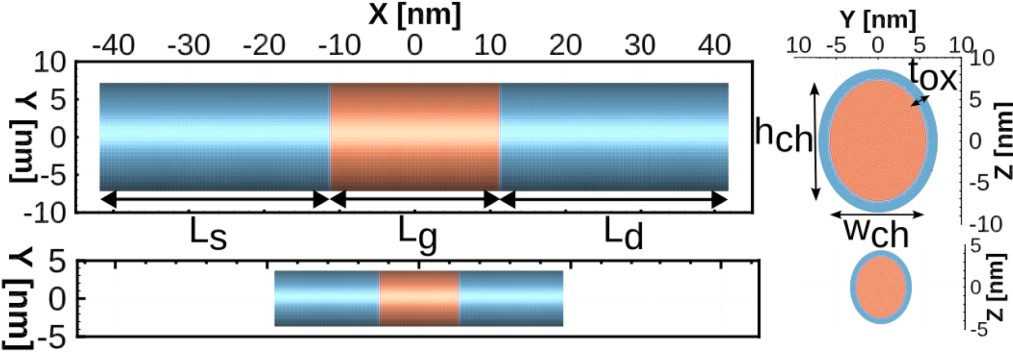

**Fig 1. 2D schematics of the 22 nm (top) and 10 nm (bottom) gate length GAA NW FETs.**

**Table 1. Dimensions, dopings and configuration parameters for the two ideal, LER-free GAA NW FETs.**

| Device | 22 nm | 10 nm |
|---|---|---|
| Gate length ($L_g$ [$nm$]) | 22 | 10 |
| Source/Drain lengths ($L_s$, $L_d$ [$nm$]) | 31.0 | 14.0 |
| Total device length ($L$ [$nm$]) | 84.0 | 38.0 |
| Semiconductor perimeter ($P_s$ [$nm$]) | 40.2 | 20.3 |
| Channel width ($w_{ch}$) [$nm$] | 11.3 | 5.7 |
| Channel height ($h_{ch}$) [$nm$] | 14.2 | 7.2 |
| Oxide thickness ($t_{ox}$) [$nm$] | 1.5 | 0.8 |
| P-type channel doping ($N_{ch}$ [$cm^{-3}$]) | $10^{15}$ | $10^{15}$ |
| N-type source/drain doping ($N_{s/d}$ [$cm^{-3}$]) | $10^{20}$ | $10^{20}$ |
| $N_{s/d}$ decay starting point ($X_m$ [$nm$]) | ±17.1 | ±7.8 |
| Slope of $N_{s/d}$ decay ($\delta$ [$nm$]) | 7.1 | 3.2 |
| Work function value (WF [$eV$]) | 4.4 | 4.4 |

work. First, a quantum-corrected drift-diffusion (DD) method, able to efficiently characterize the device behavior in the sub-threshold region. Second, a quantum-corrected ensemble Monte-Carlo (MC) approach, that produces noisy results in the sub-threshold (see Fig 3) but it is able to correctly capture non-equilibrium effects, thus being valid for calculating the device on-current ($I_{on}$). The DD approach, thanks to a careful fitting of the mobility models [18], also shows a very good agreement with the experimental data in the device on-region; however, this method has been previously demonstrated to produce inaccurate results in on-region variability studies that involve fluctuations in the device channel cross-section [19], as is the case with LER. The simulation times of one gate bias point at $V_D = 1.0$ V using either 3D quantum-

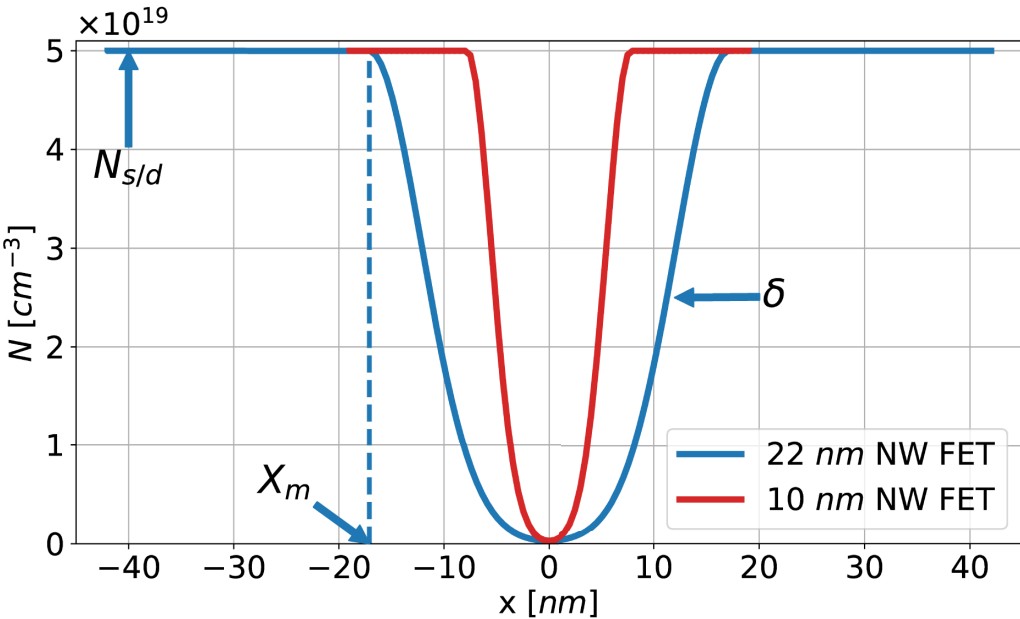

**Fig 2. Cross-section of Gaussian-like doping profile along the transport direction in the 22 nm and 10 nm gate length GAA NW FETs.**

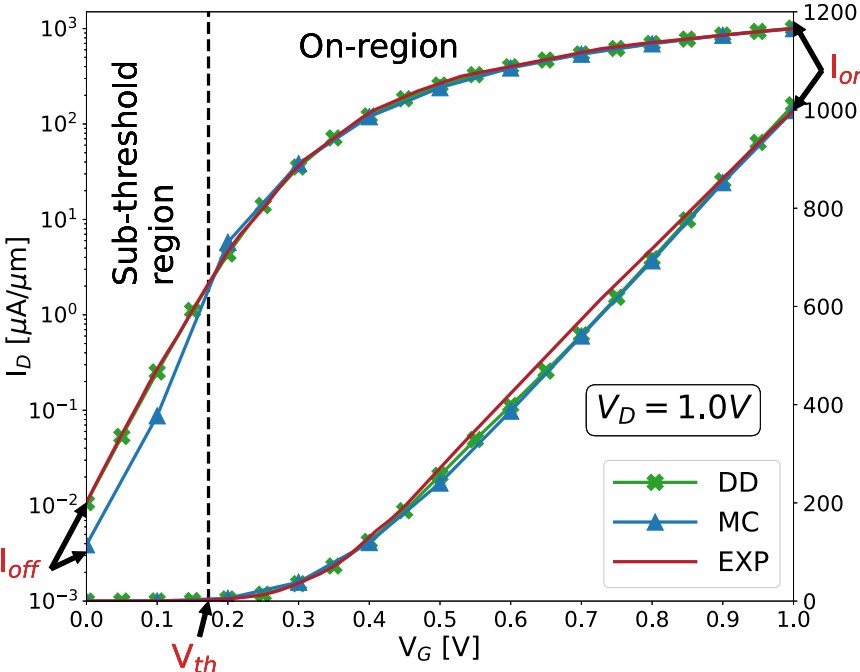

**Fig 3. Experimental (EXP) vs. simulated $I_D$-$V_G$ characteristics at $V_D$=1.0 V, on both logarithmic and linear scales, for the 22 nm gate length GAA NW FET.** Quantum-corrected drift-diffusion (DD) and Monte Carlo (MC) simulations are included for comparison. The main figures of merit (FOM) that characterize device performance, off-current ($I_{off}$), threshold voltage ($V_{th}$) and on-current ($I_{on}$) are included.

corrected DD or MC simulations are on average 1.4 hours and 80.3 hours, respectively, in an Intel i7-9700K CPU @ 3.60GHz single core for a 190 K nodes device mesh. Therefore, to save computational time and resources, we combine fast DD simulations to obtain the values of the sub-threshold region figures of merit, i.e. off-current ($I_{off}$), sub-threshold slope (*SS*) and threshold voltage ($V_{th}$), with slower MC results to extract the $I_{on}$. As indicated in Fig 3, $I_{off}$ and $I_{on}$ are calculated as the drain currents at the specific gate biases of 0.0 V and 1.0 V, respectively. $V_{th}$ is extracted via the linear extrapolation (LE) method, that defines the threshold voltage as the x-intercept of the $I_D^{0.5}-V_G$ curve linear extrapolation at its maximum first derivative point [20]. Note that, unlike the off- and on-currents, to obtain an accurate $V_{th}$ value several gate bias points need to be simulated, with average execution times of 8.4 hours. The *SS* is the slope of the linear part of the $I_D$-$V_G$ curve observed for $V_G$ values lower than $V_{th}$ (see Fig 3). Therefore, once you obtain $V_{th}$ you can also estimate *SS* without any further computation. Table 2 shows the main figures of merit for the 22 and 10 nm gate length devices, in ideal conditions, i.e. not affected by LER.

LER is a source of variability that arises from the fabrication processes since the device edges are not perfectly smooth and deviate from the ideal shape. At the current scaling level, with dimensions below 10 nm, LER can be as large as the size of the device's critical features, thus heavily impacting the transistor's performance and reliability [2]. To model LER, the edges of the nanowire in *y*-direction (see an example in Fig 4) are deformed according to the shape of a roughness profile created via the Fourier synthesis method [21]. These deformations are typically characterized by two parameters: i) the correlation length (CL), which describes the spatial correlation between deformations in the different points of the device in the *x*-direction and, ii) the root mean square (RMS) height, that establishes the amplitude of the

**Table 2. Main figures of merit that characterize the performance of the two ideal, LER-free GAA NW FETs.**

| Device | 22 nm | 10 nm |
|---|---|---|
| Supply voltage ($V_D$ [V]) | 1.0 | 0.7 |
| Off-current ($I_{off}$ [A]) | $7.28 \times 10^{-9}$ | $2.09 \times 10^{-11}$ |
| On-current ($I_{on}$ [A]) | $4.97 \times 10^{-5}$ | $3.07 \times 10^{-5}$ |
| Sub-threshold slope (SS [mV/dec]) | 85.0 | 69.7 |
| Threshold voltage ($V_{th}$ [V]) | 0.132 | 0.267 |

roughness in the $y$-direction. First, the Gaussian spectrum ($S_G$) is used to generate the roughness as follows:

$$S_G = \sqrt{\pi}(RMS)^2 e^{-k^2(CL)^2/4},\tag{1}$$

being $k$ the frequency values that are defined by the discretization in real space. Then, the spectrum $S_G$ is multiplied by an array of complex random numbers and transformed back to real space via an inverse fast Fourier transform. The applied LER is uncorrelated, i.e. the deformations will not be equal at both edges of the device, mimicking the real fabrication process. Fig 5 shows an example of a roughness profile with a CL = 10 nm and a RMS = 1 nm, which is then used to modify the device structure. Note that, the FE-based tetrahedral discretization will allow to properly capture the LER-induced deformation.

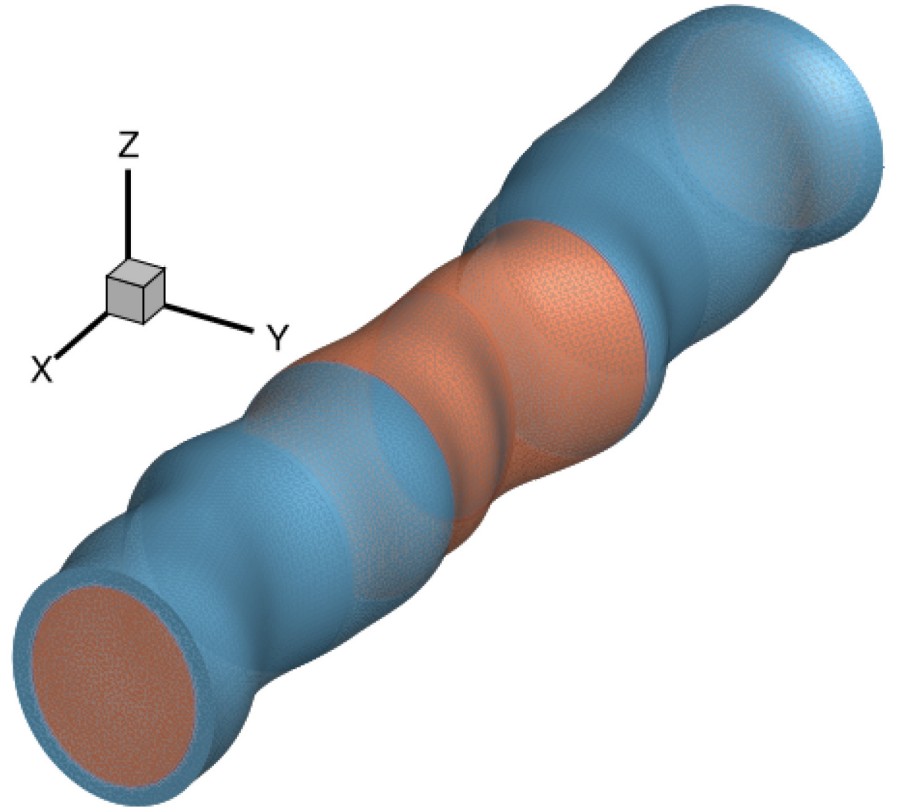

**Fig 4. Example of a 22 nm gate length GAA NW FET affected by LER.** The correlation length (CL) is 10 nm and the root mean square (RMS) height is 1 nm.

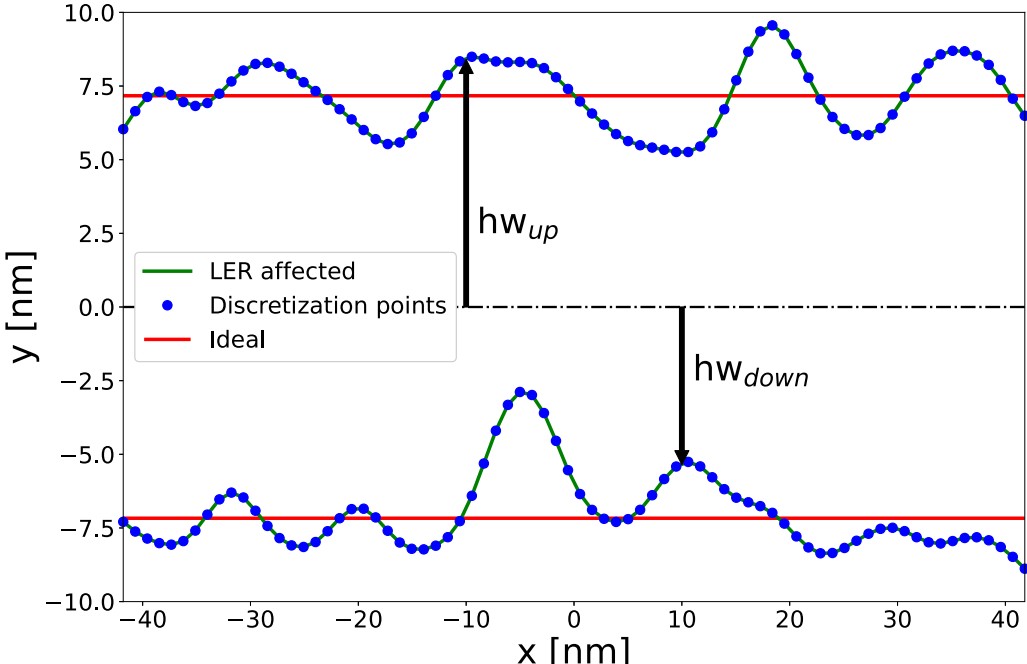

**Fig 5. Line-edge roughness deformation (CL = 10 nm, RMS = 1 nm) illustrating the effect on the 22 nm gate length device geometry.** The outline of the ideal undeformed device is included as reference.

## Machine learning modeling

Machine learning and deep learning models have been successfully applied to many research areas [22]. However, the use of such methods to deal with the most relevant transistor design challenges has only recently been initiated. As it was previously noted, the characterization of Si-based GAA NW FETs behavior requires very time-consuming simulations, especially in the case of using MC methods. For this reason, we propose a machine learning approach to predict the impact of LER on these devices with the aim of decreasing noticeably the total simulation time. In particular, to obtain the device on-current ($I_{on}$), off-current ($I_{off}$), sub-threshold slope (SS) and threshold voltage ($V_{th}$), we plan to use multi-layer perceptron (MLP) networks, which are simpler with respect to other types of neural networks but powerful enough to deliver very good results [23]. In any case, we will also compare the performance results against other well-established ML methods.

MLPs are fully connected feed-forward neural networks, which consist of three or more layers (an input and an output layer with one or more hidden layers). An example is shown in Fig 6. The input layer consists of a set of neurons (from $x_1$ to $x_n$ in the figure) representing the input features. Each neuron in the hidden layer transforms the values from the previous layer with a weighted linear summation, followed by a non-linear activation function. The output layer receives the values from the last hidden layer and transforms them into the output values. The neurons in the MLP are trained with the back propagation learning algorithm. As a result, MLPs are designed to approximate any continuous function and can solve problems which are not linearly separable for either classification or regression. In our case, we will focus on regression since the goal is to obtain the values that characterize a particular device ($I_{on}$, $I_{off}$, SS and $V_{th}$) using as input some features describing its LER deformations.

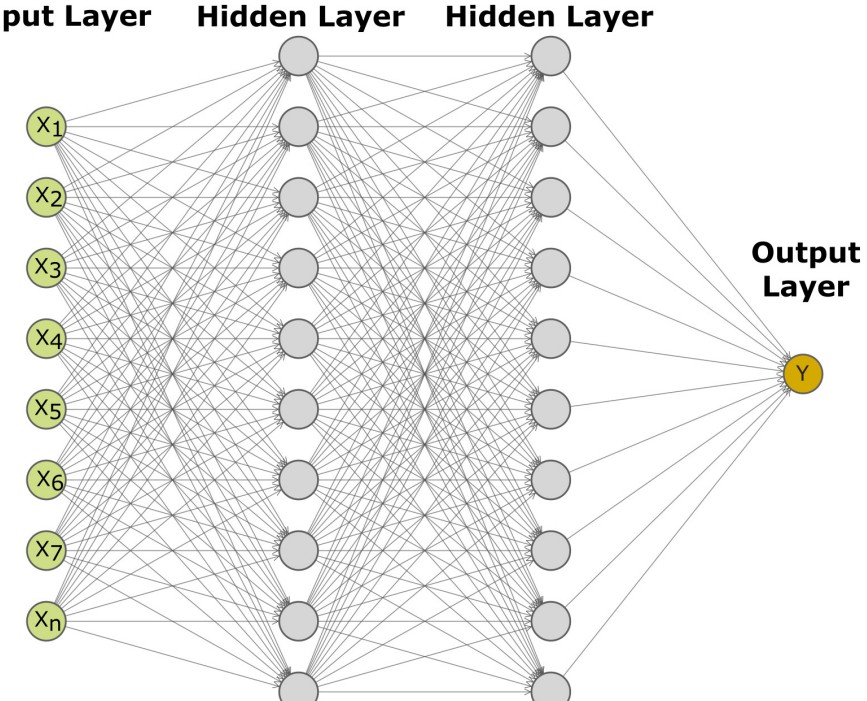

**Fig 6. Example of a multi-layer perceptron network (MLP) containing two hidden layers.**

Specifically, to generate the input features for training the neural network, the total length of the device ($x$-direction) is discretized into 400 points (see a simplified example in Fig 5), a value large enough to capture the effect of the LER deformation. At each of these points the downward vertical distance between the middle of the device ($y = 0.0$) to its edge ($y = hw_{down}$), is measured and stored, using negative values for reference. Next, the same procedure is carried out in the upward direction ($y = hw_{up}$), but now these values are considered as positive. Consequently, for each LER-affected device, there are a total of 800 input values that characterize its deformation. From now on, we refer to these points as the LER profile of the device.

## Results and discussion

### Datasets and experimental setup

As mentioned before, in this work we use two GAA NW FETs that differ both in their physical dimensions and in other configuration parameters, as shown in Table 1. LER deformations are then applied to these ideal non-deformed devices considering three RMS heights (0.4, 0.6 and 1.0 nm), and four CL values (10, 15, 20 and 30 nm), generating 1,000 different device configurations (LER profiles) for each combination of RMS height and CL. For each LER-affected device, as mentioned in the Methodology section, we run a DD simulation to extract the subthreshold region figures of merit, and a Monte Carlo simulation, to obtain the $I_{on}$. Note that, although the material properties of the devices under study are exactly the same, their physical dimensions will differ due to LER. Therefore, for a particular device configuration, it may occur that one simulation methodology is able to reach convergence while the other one fails to do so, providing a *Null* output in the corresponding figure of merit, which is then disregarded for the study. It is worth mentioning that the longer the RMS height, the larger the

influence of LER on the device performance [24]. The impact of LER also tends to grow for increasing CLs, although it reaches a plateau at CL values similar to the device gate length [25]. For these reasons, for both gate length devices, we have combined the extreme values of CL (10 and 30 nm) and RMS (0.4 and 1.0 nm) to generate the training datasets, and we have used two intermediate values of CL and RMS, i.e. CL = 15 nm, RMS = 0.6 and CL = 20 nm, RMS = 0.4, to generate the test datasets. As a consequence, the training and test datasets for each gate length device contain 4,000 and 2,000 examples, respectively.

All the codes used for this work have been implemented using Python 3.9 and the Scikit-learn library (v1.2.1). The main features of the MLP network used in the experiments are detailed in Table 3. Note that the number of hidden layers and their sizes (hyperparameters) were previously obtained using a grid search. To train the MLP model, LBFGS (Limited-memory Broyden-Fletcher-Goldfarb-Shanno algorithm) was adopted because of the relatively small size of the training data set (4,000 examples) [26]. It uses the square error as loss function. The response variables $I_{off}$ and $I_{on}$ are scaled using the logarithmic function as a pre-processing stage, so the hyperbolic tangent (tanh) is the most suitable activation function because it can work for positive as well as negative input values [12]. On the other hand, all the experiments were conducted on a server with one Intel Core i7-9700K CPU @ 3.60GHz and 128 GB of RAM memory.

## Performance results

To evaluate and compare the different machine learning approaches, we have considered two performance metrics:

Coefficient of determination ($R^2$): It is a measure that provides information about the goodness of fit of a model. In the context of regression it is a statistical measure of how well the regression line approximates the actual data and therefore a measure of how well unseen examples are likely to be predicted by the model. If $\hat{y}_i$ is the predicted value of the *i-th* example and $y_i$ is the corresponding true value for total $n$ examples, the estimated $R^2$ is defined as:

$$R^2 = 1 - \frac{\sum_{i=1}^{n} (y_i - \hat{y}_i)^2}{\sum_{i=1}^{n} (y_i - \bar{y})^2} \tag{2}$$

where $\bar{y} = \frac{1}{n} \sum_{i=1}^{n} y_i$. The best possible score is 1 and it can be negative.

**Table 3. Main characteristics of the MLP network considered in this work.** Note that the solver refers to the algorithm or method used to solve the optimization problem involved in training the regressor. L2 regularization adds a penalty term to the loss function during training to prevent overfitting.

| Parameter | Value |
|---|---|
| Hidden layers | 3 |
| Neurons per hidden layer | 80 |
| Solver | LBFGS |
| Activation function | tanh |
| Max. Iterations | 2,000 |
| L2 regularization | 0.1 |
| Input features | 800 (LER profile) |

Root Mean Squared Error (*RMSE*): It is the most common evaluation metric for regression models. It is the square root of the mean squared error (*MSE*):

$$\sqrt{\left(\frac{1}{n}\right)\sum_{i=1}^{n}(y_i - \hat{y}_i)^2} \tag{3}$$

The calculated value is in the same unit as the required output variable. Lower values are better.

Figs 7 and 8 show the performance of our trained MLP model by comparing predicted and actual values of the four figures of merit ($I_{off}$, $I_{on}$, *SS* and $V_{th}$) for the 22 nm and 10 nm LER-affected devices, respectively. It can be observed that the predictions are noticeably accurate in all cases, also including the extreme values. It demonstrates that it is possible to predict the

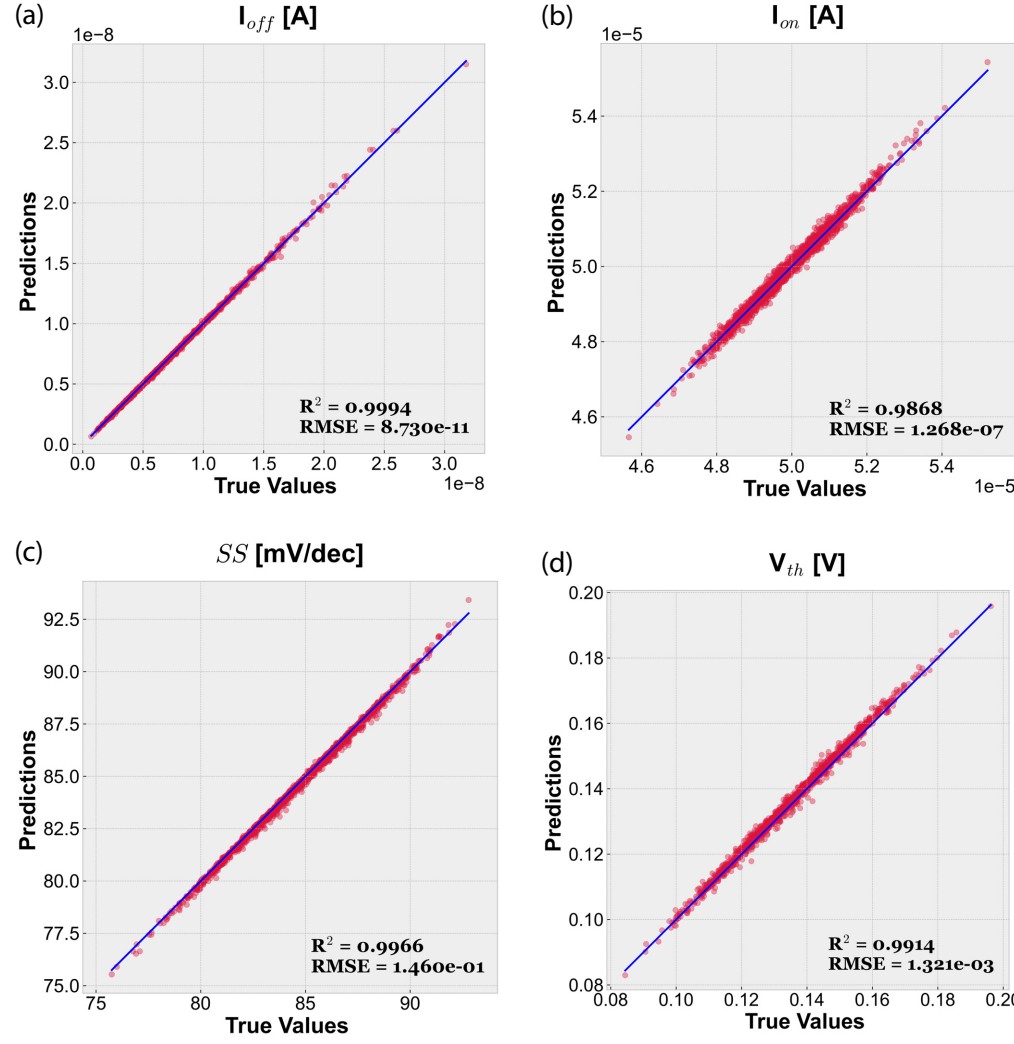

**Fig 7. Predicted and actual values for the considered figures of merit using our test dataset (LER-affected 22 nm gate length GAA NW FETs).**

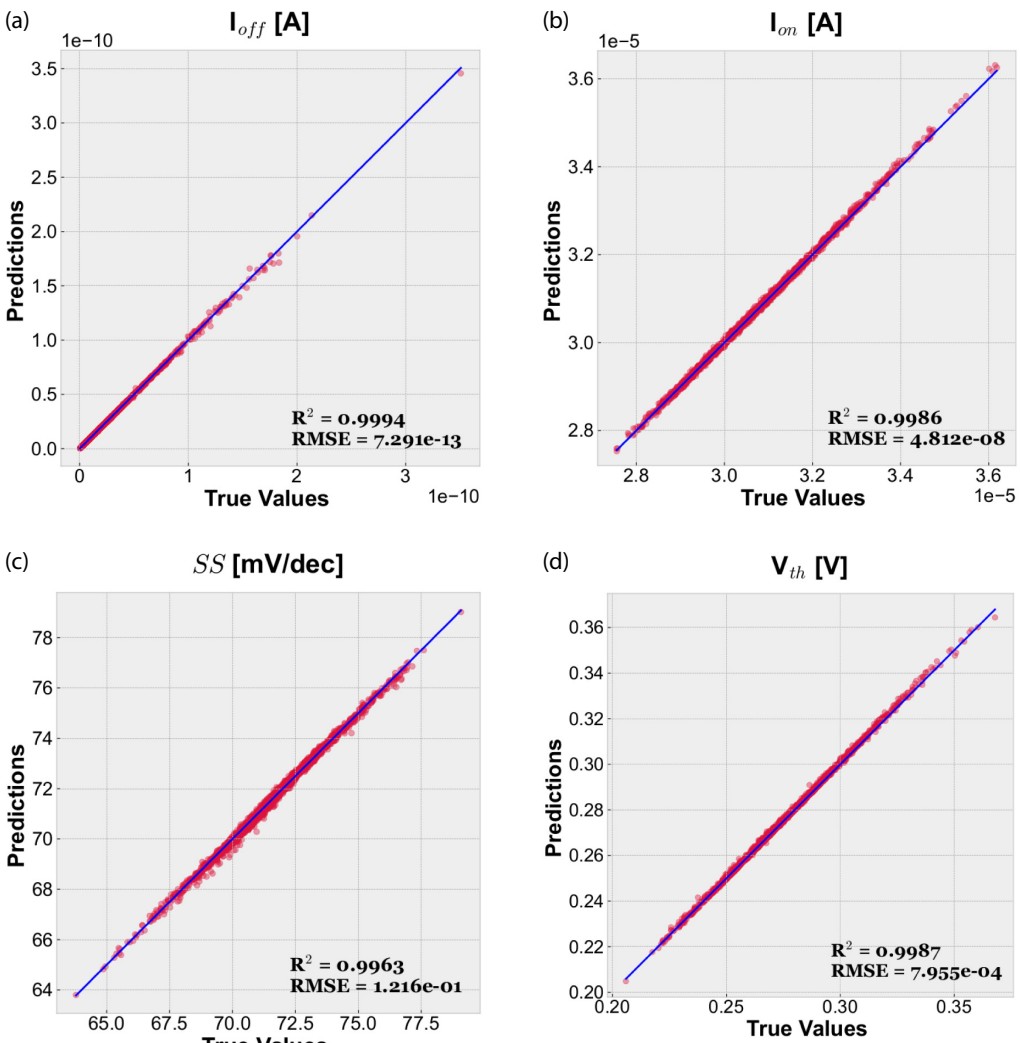

**Fig 8. Predicted and actual values for the considered figures of merit using our test dataset (LER-affected 10 nm gate length GAA NW FETs).**

behavior of devices affected by intermediate values of CL and RMS using as input of the MLP only information obtained from their extreme values. In particular, Table 4 summarizes the $R^2$ and *RMSE* values achieved by our models. Performance metrics confirm that predictions are excellent, reaching $R^2$ values up to 0.9994. Note that the worst case, $I_{on}$ for 22 nm devices, is still very good with a coefficient of determination of 0.9868. *RMSE* is always very low, being at least two orders of magnitude lower than the actual values of the considered response variable. See in Table 2 the different figures of merit reference values for the ideal GAA NW FETs.

The above results were obtained using the complete training datasets to feed the MLP networks. Next we will evaluate the impact of the input data size on the training process. With this goal in mind, $R^2$ and *RMSE* performance metrics were computed for MLP networks trained using only a fraction of the input dataset ranging from 0.1 (10% of the dataset) to 1 (complete dataset). Results when considering the 22 nm devices are displayed in Fig 9. It can be observed that even for small percentages of the input dataset, metrics for all the response variables are quite good. For instance, $R^2$ values range from 0.9642 ($V_{th}$) to 0.9919 ($I_{off}$) using

**Table 4. Performance metrics ($R^2$ and *RMSE*) of our MLP-based regression models.**

| Figs. of merit | 22 nm | | 10 nm | |
|---|---|---|---|---|
| | $R^2$ | *RMSE* | $R^2$ | *RMSE* |
| $I_{off}$ [A] | 0.9994 | $8.730 \times 10^{-11}$ | 0.9994 | $7.291 \times 10^{-13}$ |
| $I_{on}$ [A] | 0.9868 | $1.268 \times 10^{-7}$ | 0.9986 | $4.812 \times 10^{-8}$ |
| SS [mV/dec] | 0.9966 | $1.460 \times 10^{-1}$ | 0.9963 | $1.216 \times 10^{-1}$ |
| $V_{th}$ [V] | 0.9914 | $1.321 \times 10^{-3}$ | 0.9987 | $7.955 \times 10^{-4}$ |

only 10% of the training dataset. The response variable that benefits the most from the increase in the number of input examples is $V_{th}$ (bottom right figure). On the other hand, as expected, *RMSE* tends to decrease when adding more examples to the training dataset. It is worth noting that the behavior for the 10 nm devices is very similar. Therefore, our approach is capable of successfully predicting the values of the figures of merit even using a reduced training dataset.

Training our MLP models is extremely fast. In particular, it only takes on average from 50.9 to 162.6 seconds depending on the considered response variable and gate length. It means that, for example, computing $I_{on}$ from our trained model for a particular 22 nm device is about 2,370 × faster than using a MC simulation (80.3 hours, as was explained in the Methodology section). Note that times to generate the training data are not included in these experiments.

**Comparison with other machine learning methods.** In addition to MLPs, we can find in the literature many regression methods. To demonstrate the benefits of our approach, a comparison with some of the most successful regression techniques was carried out. In particular:

- Decision Tree (DT) regression [27]: It creates a tree-based structure that predicts the value of a target variable by learning simple decision rules inferred from the data features. A

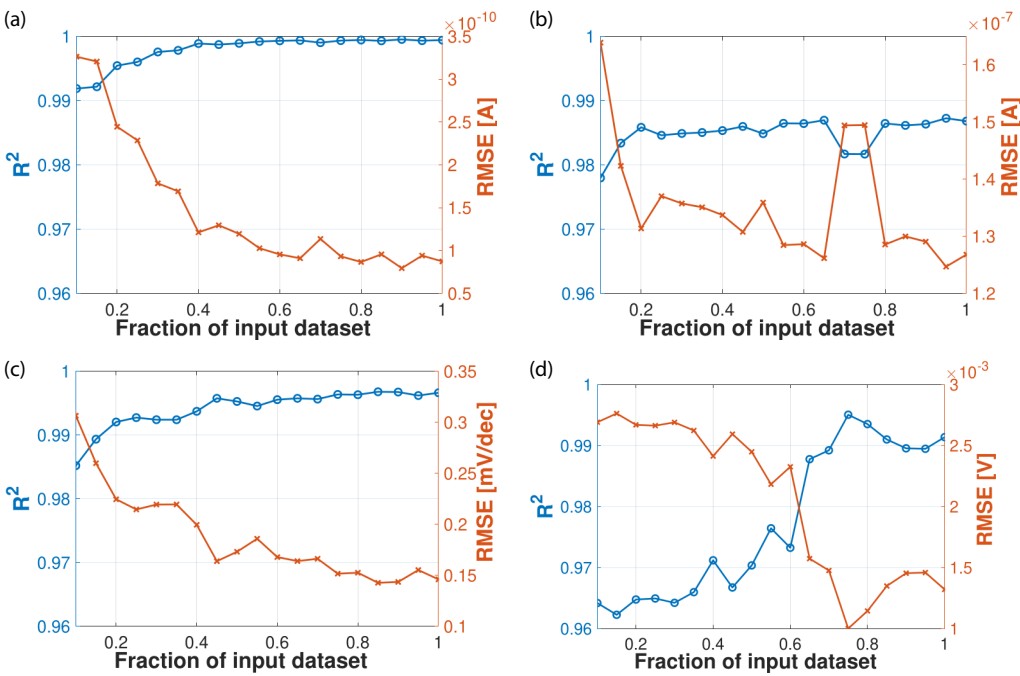

**Fig 9. Impact of the training data size on the performance of the MLP models (LER-affected 22 nm gate length GAA NW FETs).** (a) $I_{off}$. (b) $I_{on}$. (c) SS. (d) $V_{th}$.

regression tree is built using a binary recursive partitioning process. Initially, all the training examples are grouped into the same partition. The algorithm then begins allocating the data into the first two partitions or branches, using every possible binary split on every feature. The algorithm selects the split that minimizes the sum of the squared deviations from the mean in the two separate partitions. This splitting rule is then applied to each of the new branches. To predict a response, the decisions in the tree should be followed from the root (beginning) node down to a leaf node. The leaf node contains the response.

- Random Forest (RF) regression [28]: It is a supervised learning algorithm that uses ensemble learning methods for regression. In particular, it uses the combination of multiple random decision trees, each trained on a subset of data. The use of multiple trees gives stability to the algorithm and reduces variance.

- Support Vector Machine (SVM) regression [29]: It is one of the most successful and well studied methods for regression. SVM regression is considered a nonparametric technique because it relies on kernel functions. One of the main advantages of SVM regression is that its computational complexity does not depend on the dimensionality of the input space.

Table 5 shows the performance metrics using different machine learning regression techniques for the 22 nm and 10 nm LER-affected devices. For all the response variables, the best performer is always our proposal based on MLP networks. Differences with respect to the other methods are noticeable. Regardless, RF obtains decent results for all the studied cases. We must highlight that RF training times are always higher than those observed for MLP. DT and SVM are faster to train, but their results are poor and very irregular.

**Transfer learning.** Machine learning methods, especially those related to (deep) neural networks, require big datasets to successfully train the models. There are scenarios where

**Table 5. Performance metrics obtained by different machine learning techniques for regression.**

| FOM | Regressor | 22 nm | | 10 nm | |
|---|---|---|---|---|---|
| | | $R^2$ | RMSE | $R^2$ | RMSE |
| $I_{off}$ [A] | **MLP** | **0.9994** | **$8.730 \times 10^{-11}$** | **0.9994** | **$7.291 \times 10^{-13}$** |
| | DT | 0.4950 | $2.571 \times 10^{-9}$ | 0.7866 | $1.347 \times 10^{-11}$ |
| | RF | 0.9360 | $9.156 \times 10^{-10}$ | 0.9601 | $5.820 \times 10^{-12}$ |
| | SVM | 0.9736 | $5.874 \times 10^{-10}$ | 0.9802 | $4.109 \times 10^{-12}$ |
| $I_{on}$ [A] | **MLP** | **0.9868** | **$1.268 \times 10^{-7}$** | **0.9986** | **$4.812 \times 10^{-8}$** |
| | DT | 0.8501 | $4.274 \times 10^{-7}$ | 0.9427 | $3.081 \times 10^{-7}$ |
| | RF | 0.9615 | $2.166 \times 10^{-7}$ | 0.9889 | $1.352 \times 10^{-7}$ |
| | SVM | 0.3997 | $8.554 \times 10^{-7}$ | 0.7380 | $6.589 \times 10^{-7}$ |
| SS [mV/dec] | **MLP** | **0.9966** | **0.146** | **0.9963** | **0.122** |
| | DT | 0.6697 | 1.448 | 0.8644 | 0.735 |
| | RF | 0.9446 | 0.5932 | 0.9761 | 0.309 |
| | SVM | 0.7884 | 1.159 | 0.3203 | 1.646 |
| $V_{th}$ [V] | **MLP** | **0.9914** | **$1.321 \times 10^{-3}$** | **0.9987** | **$7.955 \times 10^{-4}$** |
| | DT | 0.7392 | $7.262 \times 10^{-3}$ | 0.9181 | $6.395 \times 10^{-3}$ |
| | RF | 0.9501 | $3.175 \times 10^{-3}$ | 0.9869 | $2.553 \times 10^{-3}$ |
| | SVM | 0.5932 | $9.071 \times 10^{-3}$ | -0.2088 | $2.457 \times 10^{-2}$ |

Bold numbers highlight the best performer for each figure of merit.

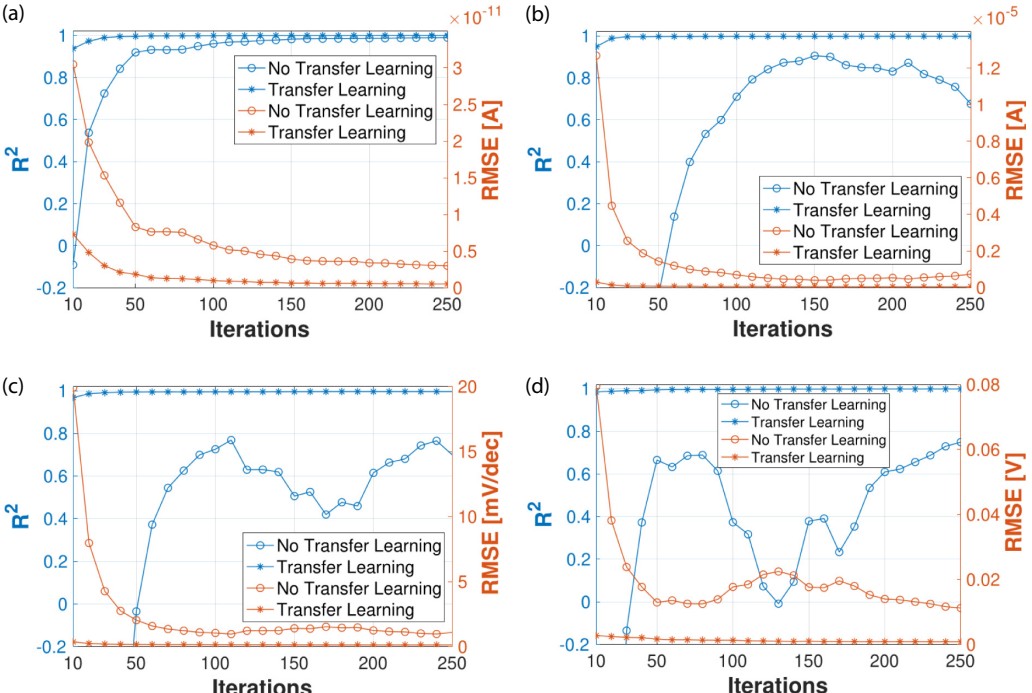

**Fig 10. Performance metrics using a transfer learning approach and training the networks from scratch (i.e., without transfer learning) to predict the figures of merit of the 10 nm gate length devices.** (a) $I_{off}$. (b) $I_{on}$. (c) $SS$. (d) $V_{th}$.

training data is expensive or difficult to collect. In our case, for example, a single MC simulation takes tenths of hours on a standard server. This is where transfer learning comes in.

Transfer Learning (TL) refers to a technique for predictive modeling on a different but somewhat related problem that can then be reused partially or completely to speed up training and/or improve a model's performance on the problem of interest. In the context of neural networks, this means reusing the weights of one or more layers of a pre-trained network model in a new model and keeping the weights fixed, adjusting them or adapting them completely when training the model.

Next, we will apply a transfer learning approach to predict the figures of merit of the 10 nm devices using as starting point the trained models used for the 22 nm devices. It means that we will retain the values of the model's trainable parameters from the previous model (22 nm devices) and use those initially instead of starting a training process from scratch. First, we want to demonstrate that this way the training process will be faster. That is, the number of iterations required to successfully train the networks is significantly lower. A comparison between training the networks from scratch or using the transfer learning approach is shown in Fig 10. The graphs display the results of the training process in terms of the evolution of $R^2$ and $RMSE$ when using different numbers of iterations. It can be observed that the transfer learning method works since $R^2$ and $RMSE$ quickly reach values very close to the maximum and minimum, respectively. For instance, $R^2$ and $RMSE$ are 0.9986 and $4.812 \times 10^{-8}$ A when training the network from scratch to predict $I_{on}$ for the 10 nm LER-affected devices (see Table 4). In that case, the number of iterations was 2,000. Using transfer learning, the corresponding values after only 100 iterations are 0.9981 and $5.635 \times 10^{-8}$ A, which are almost identical to our best ones (top right Fig 10). Note that, if we train the MLP model without transfer

**Table 6. Performance metrics obtained by our transfer learning approach when using a small fraction of the training dataset to predict the figures of merit of the 10 nm gate length devices.**

| FOM | Percentage of training dataset | | | | | |
|---|---|---|---|---|---|---|
| | 1% | | 5% | | 10% | |
| | $R^2$ | $RMSE$ | $R^2$ | $RMSE$ | $R^2$ | $RMSE$ |
| $I_{off}$ [A] | 0.9256 | $7.955 \times 10^{-12}$ | 0.9912 | $2.742 \times 10^{-12}$ | 0.9959 | $1.864 \times 10^{-12}$ |
| $I_{on}$ [A] | 0.8603 | $4.812 \times 10^{-7}$ | 0.9930 | $1.077 \times 10^{-7}$ | 0.9958 | $8.377 \times 10^{-8}$ |
| $SS$ [mV/dec] | 0.8261 | $8.328 \times 10^{-1}$ | 0.9784 | $2.933 \times 10^{-1}$ | 0.9888 | $2.111 \times 10^{-1}$ |
| $V_{th}$ [V] | 0.7019 | $1.220 \times 10^{-2}$ | 0.9816 | $3.034 \times 10^{-3}$ | 0.9859 | $2.654 \times 10^{-3}$ |

learning instead, after 100 iterations, $R^2$ and $RMSE$ would be 0.7098 and $6.934 \times 10^{-7}$ A, very far from our best results. At the same time, reducing the iterations to converge also has a big impact on the training times. In this way, considering 100 iterations, we reduce to less than 5 seconds the time required to train our MLP-based models in order to predict the response variables. In other words, computing the figures of merit for the 10 nm devices when using transfer learning is about 57,800× and 1,000× faster than MC and DD simulations, respectively (see Methodology section).

As commented above, another important advantage of transfer learning is the reduction of the required input data to train the models, which is especially relevant in cases where training data is costly or difficult to collect. Table 6 shows the regression performance metrics obtained by our transfer learning approach when using a small fraction of the training dataset to predict the response variables of the 10 nm devices. Results confirm the benefits of our methodology where good predictions are achieved even when using a small percentage of the training dataset. For example, $R^2$ is always above 0.985 using 10% of the input examples (i.e. only 400 LER profiles).

Therefore, we can conclude that transfer learning is a good solution to speed up the training process and also to reduce noticeably the required training dataset size. This technique could aid in the design of variability-resistant device architectures since it could allow quick and simple testing of the impact of different device features (e.g. gate length, cross-section dimensions) on an LER-affected transistor's performance.

**Impact on the environment: Carbon emissions.** As we pointed out, our approach reduces noticeably the computing time to calculate the figures of merit for a particular device. Next, we demonstrate that it also has a strong impact on the environment, causing a reduction in the carbon footprint. To estimate the carbon emissions we follow the methodology presented in Lacoste *et al.* [30]. In particular, the estimated carbon emissions in grams are derived using the following expression:

$$\text{eq. grams of } CO_2 = \frac{t \cdot C_e \cdot W_{\text{cpu}}}{1000} \tag{4}$$

where $t$ is the equivalent CPU-hours of computation, $C_e = 341$ is the carbon efficiency coefficient of the grid (measured in grams of $CO_2$eq/kWh) and $W_{\text{cpu}}$ is the Thermal Design Power of the CPU in watts (95 W in our case). Note that the carbon efficiency data for our region was taken from Moro and Lonza [31]. We use the corresponding CPU-hours required to compute the figures of merit by means of simulations (DD and MC) and training the MLP models (our approach).

Carbon emissions are shown in Table 7. We compare the calculation of the figures of merit for an LER-affected 10 nm device using simulations (DD and MC) and our proposal based on

**Table 7. Carbon emissions in grams of $CO_2$ to compute the figures of merit of only one LER-affected 10 nm gate length GAA NW FET using simulations and our MLP-based approach with and without transfer learning.**

| Figs. of merit | $CO_2$ emissions (g) | | |
|---|---|---|---|
| | Simulations | Training (no TL) | Training (TL, 100 iters.) |
| $I_{off}$ | 45.35 (DD) | 0.90 | 0.04 |
| $I_{on}$ | 2,601.32 (MC) | 0.84 | 0.05 |
| $SS$ | 272.12 (DD) | 0.53 | 0.04 |
| $V_{th}$ | 272.12 (DD) | 0.46 | 0.05 |

training MLP networks. It is important to highlight that, unlike the simulations procedure, the carbon footprint of the training process should be paid only once, because the same trained network can be reused for different configurations (LER profiles) of devices with equal gate length. As a result, for instance, our proposal reduces the emissions from 2.6 kg to just 0.84 g of $CO_2$ for the calculation of $I_{on}$ when training the MLP network from scratch. However, if 1,000 configurations are considered, the carbon emissions caused by the MC simulations will increase up to 2.6 tons of $CO_2$, while our method does not require additional training. If the transfer learning method is used instead, the carbon footprint is dramatically reduced to only 0.05 g of $CO_2$.

## Conclusions

The digital world we live in would have not been possible without the continuous advance of the semiconductor industry. In this context, the use of advanced simulation tools (TCAD) to evaluate new semiconductor device architectures and assess their robustness is crucial for both the semiconductor industry and academic research. However, with the current device's critical dimensions deep into the nanometer regime, the computational cost of some TCAD studies can be prohibitive. Therefore, the introduction of less computationally-demanding methods is needed to deal with this problem. Here, we have demonstrated the advantages of using machine learning techniques to assess the effect of the line edge roughness-induced variability on gate-all-around nanowire (GAA NW) FETs. The impact of LER on four different figures of merit (off-current, threshold voltage, sub-threshold slope and on-current) has been predicted for two different GAA NW FETs, a 22 nm gate length device and a scaled-down version, with a 10 nm gate length. The MLP networks have achieved the best performance metrics ($R^2$ and *RMSE* values), when compared to well-known regression methods (DT, RF and SVM), with $R^2 \sim 0.99$ for the two devices and the four analyzed figures of merit. Finally, we demonstrate that MLP networks can dramatically decrease variability studies computational effort, which can be diminished even further by using transfer learning techniques, achieving $R^2 > 0.985$ when using only a 10% of the input samples, and producing as little as 0.18 g of $CO_2$ emissions (when computing the four studied figures of merit), a value several orders of magnitude lower than that of TCAD studies. Finally, it is worth mentioning that the MLP architecture could also be applied (with an adequate calibration of the network hyperparameters and weights) to other relevant sources of variability affecting semiconductor devices, such as metal grain granularity, gate-edge roughness or random discrete dopants.

## Author Contributions

**Conceptualization:** Juan C. Pichel.

**Funding acquisition:** Antonio García-Loureiro.

**Investigation:** Antonio García-Loureiro.

**Software:** Antonio García-Loureiro, Natalia Seoane, Julián G. Fernández, Enrique Comesaña, Juan C. Pichel.

**Supervision:** Antonio García-Loureiro, Juan C. Pichel.

**Writing – original draft:** Natalia Seoane, Juan C. Pichel.

**Writing – review & editing:** Natalia Seoane, Juan C. Pichel.

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
