## [Decision Letter · Decision Letter 0]

18 Jun 2023

PONE-D-23-15843A machine learning approach to model the impact of line edge roughness on gate-all-around nanowire FETs while reducing the carbon footprintPLOS ONE

Dear Dr. Loureiro,

Thank you for submitting your manuscript to PLOS ONE. After careful consideration, we feel that it has merit but does not fully meet PLOS ONE’s publication criteria as it currently stands. Therefore, we invite you to submit a revised version of the manuscript that addresses the points raised during the review process.

We look forward to receiving your revised manuscript.

Kind regards,

Talib Al-Ameri, Ph.D

Academic Editor

PLOS ONE

Journal Requirements:

"Work supported by the Spanish MICINN, Xunta de Galicia, and FEDER Funds under Grants RYC-2017-23312, PID2019-104834GB-I00, PLEC2021-007662, ED431F 2020/008 and ED431C 2022/16."

"Work supported by the Spanish MICINN, Xunta de Galicia, and FEDER Funds under 332

Grants RYC-2017-23312, PID2019-104834GB-I00, PLEC2021-007662, ED431F 2020/008 333

and ED431C 2022/16."

"Work supported by the Spanish MICINN, Xunta de Galicia, and FEDER Funds under Grants RYC-2017-23312, PID2019-104834GB-I00, PLEC2021-007662, ED431F 2020/008 and ED431C 2022/16."

Reviewers' comments:

Reviewer's Responses to Questions

**Comments to the Author**

1. Is the manuscript technically sound, and do the data support the conclusions?

Reviewer #1: Yes

2. Has the statistical analysis been performed appropriately and rigorously? 

Reviewer #1: Yes

3. Have the authors made all data underlying the findings in their manuscript fully available?

Reviewer #1: Yes

4. Is the manuscript presented in an intelligible fashion and written in standard English?

Reviewer #1: Yes

5. Review Comments to the Author

Reviewer #1: Review PlusOne PONE-D-23-15843

A machine learning approach to model the impact of line edge roughness on gate-all- around nanowire FETs while reducing the carbon footprint

The paper deals with a highly efficient optmisation of the calculation of intrinsic variability in the nanoscale multigate transistors by using a machine learning algorithm. The usage of the machine learning approach will undoubtedly make the calculation of variability much more efficient when compared to a power approach when device simulations are run repeatedly with different sets of parameters affected by the variability sources. The paper will be of a great interest to readers and I would support it publication when following comments are addressed:

1) p. 1. & 13, line 10 & 308: The right expression has to be used instead of expression ‘...advanced simulation techniques…’ for the abbreviation of TCAD.

2) p. 2, line 21: In the sentence ‘...is relevant to apply alternative techniques, such as machine learning (ML), …’ , a reference to the machine learning technique is missing. Additionally, why the ML is viewed as an alternative technique to traditional TCAD techniques? The ML is a type of computational statistic using learning from data sets while the typical TCAD techniques will be physically based simulation techniques.

3) p. 3: A schematic of doping profile illustrating significance of doping parameters as N_s/d and X_m, and delta should be added either to Fig. 1 or as a new figure.

4) p. 3: The assumed workfunction should be mentioned for the simulations of ideal 10 nm and 22 nm gate NW FETs.

5) p. 5, lie 102 – 117: The introduction of the multi-layer perceptron (MLP) networks has no reference to the method. Is this method invented by authors? If so, theoretical description of the method has to be given. If not, a reference must be given.

6) p. 6, line 134: A number is missing in ‘...For each LER-affected device, as mentioned in Section ,’.

7) p. 6, line 153: The abbreviation LBFGS must be spelled out.

8) p. 7, Table 3: The table caption should explain in details (ideally each of them) what are the parameters listed in the table for a reader to have a better understanding.

9) p. 7, line 174: Typo in FigS.

10) p. 8, Fig. : The text in the figure caption ‘(LER-affected 22 nm devices).’ does not refer well to the studied NW FET, what are the 22 nm devices?

11) p. 8: Does the times required to train the MLP model include the time to produce the data for the training? This should be clarified.

12) p. 11, line 268: A number is missing in (see Section ).

13) p. 11: How exactly is the Transfer Learning performed? More details how the TL when predicting the variability for the 10 nm gate NW FET is done when using the machine learning outcome for the 22 nm gate NW FET. For example, are hidden layers modified in some way, etc.?

14) A few comments could be added into Conclusions about applicability of the developed MLP network for the other sources of variability due to metal grains in the gate, line edge of the gate, and random dopands.

6. PLOS authors have the option to publish the peer review history of their article (what does this mean?). If published, this will include your full peer review and any attached files.

Reviewer #1: **Yes: **Karol Kalna

---

## [Author Response · Author response to Decision Letter 0]

4 Jul 2023

We would like to thank the reviewer for their constructive comments that will undoubtedly improve the quality of the manuscript. Next, we address each comment hoping that our answers will be to the reviewer's satisfaction.

Reply to Reviewer’s Comments to the Author

The paper deals with a highly efficient optimisation of the calculation of intrinsic variability in the nanoscale multigate transistors by using a machine learning algorithm. The usage of the machine learning approach will undoubtedly make the calculation of variability much more efficient when compared to a power approach when device simulations are run repeatedly with different sets of parameters affected by the variability sources. The paper will be of a great interest to readers and I would support it publication when following comments are addressed:

1) p. 1. & 13, line 10 & 308: The right expression has to be used instead of expression ‘...advanced simulation techniques…’ for the abbreviation of TCAD.

Reply: “Advanced simulations techniques” has been changed by the right abbreviation of TCAD, i.e. “Technology Computer-Aided Design (TCAD)”.

2) p. 2, line 21: In the sentence ‘...is relevant to apply alternative techniques, such as machine learning (ML), …’ , a reference to the machine learning technique is missing. Additionally, why the ML is viewed as an alternative technique to traditional TCAD techniques? The ML is a type of computational statistic using learning from data sets while the typical TCAD techniques will be physically based simulation techniques.

Reply: ML learning techniques have been applied, for some years now, for the regression and prediction of the behavior of different quantities in physical systems. The reviewer is right, this type of techniques do not replace traditional simulation, since the latter provides more complete information on the systems being analyzed. However, due to the increasing complexity of the systems, the availability of regressors that collect complex correlations between physical variables and assist in the prediction of trends in their behavior allows more refined or oriented simulations for the analysis of optimal configurations or of a certain interest. In our case, we are evaluating the feasibility of using machine learning techniques as regression tools for figures of merit of nanometer transistors. In these devices, the correlations between the different variables no longer respond to classical parameterizations and are complex to visualize and analyze. Therefore, we believe that the use of neural networks, among other regression techniques, can provide relevant information that saves significant computational time, minimizing the computational needs for the studies described in this article. In the work presented in this paper, we demonstrate that different machine learning techniques allow obtaining regression models suitable for this type of devices and, in particular, the use of perceptrons, results in the most efficient method in terms of computation time versus the quality of the predictions obtained.

Therefore, we have replaced in the manuscript the expression “...is relevant to apply alternative techniques, such as machine learning (ML)” by “is relevant to apply complementary techniques, such as machine learning (ML)”.

In addition, the following references related to ML have been added to the manuscript: 

1. Machine-Learning-Based Compact Modeling for Sub-3-nm-Node Emerging Transistors. https://doi.org/10.3390/electronics11172761

2. Simulator acceleration and inverse design of fin field-effect transistors using machine learning. https://doi.org/10.1038/s41598-022-05111-3

3) p. 3: A schematic of doping profile illustrating significance of doping parameters as N_s/d and X_m, and delta should be added either to Fig. 1 or as a new figure.

Reply: The reviewer is right, that kind of figure will help to understand the significance of the relevant doping parameters. Therefore, we have included a new Figure (Fig. 2 in the manuscript) that presents the schematic doping profile for both 22 nm and 10 nm GAA NW FETs, and we have referred to it as follows: 

“These devices have an uniform p-type doping in the semiconductor channel and a n-type Gaussian doping in the source/drain (s/d) regions, that is fixed to Ns/d from the s/d contacts till a point (Xm) nearby the gate region, where the doping exponentially decays (with a slope δ), as shown in Fig. 2.”

4) p. 3: The assumed workfunction should be mentioned for the simulations of ideal 10 nm and 22 nm gate NW FETs.

Reply: The work function values used for both NW FETs have been included in Table 1.

5) p. 5, line 102 – 117: The introduction of the multi-layer perceptron (MLP) networks has no reference to the method. Is this method invented by authors? If so, theoretical description of the method has to be given. If not, a reference must be given.

Reply: The revised manuscript now includes a new reference [23] (Haykin S. Neural Networks and Learning Machines. Prentice Hall; 2009).

6) p. 6, line 134: A number is missing in ‘...For each LER-affected device, as mentioned in Section ,’.

Reply: Thank you for noticing this mistake. The sentence has been modified to “For each LER-affected device, as mentioned in the Methodology section, we run a …”.

7) p. 6, line 153: The abbreviation LBFGS must be spelled out.

Reply: The abbreviation has been spelled out as follows: “LBFGS (Limited-memory Broyden-Fletcher-Goldfarb-Shanno algorithm)”.

8) p. 7, Table 3: The table caption should explain in details (ideally each of them) what are the parameters listed in the table for a reader to have a better understanding.

Reply: Following the suggestion of the reviewer, now the caption of the table clarifies some of the parameters. The caption is now as follows:

“Table 3. Main characteristics of the MLP network considered in this work. Note that the solver refers to the algorithm or method used to solve the optimization problem involved in training the regressor. L2 regularization adds a penalty term to the loss function during training to prevent overfitting.”

9) p. 7, line 174: Typo in FigS.

Reply: The capital “S” has been replaced by “s”.

10) p. 8, Fig. : The text in the figure caption ‘(LER-affected 22 nm devices).’ does not refer well to the studied NW FET, what are the 22 nm devices?

Reply: Captions of Figs. 7, 8 and 9 have been modified to indicate the studied devices as follows:

“Fig 7. Predicted and actual values for the considered figures of merit using our test dataset (LER-affected 22 nm gate length GAA NW FETs).”

“Fig 8. Predicted and actual values for the considered figures of merit using our test dataset (LER-affected 10 nm gate length GAA NW FETs).”

“Fig 9. Impact of the training data size on the performance of the MLP models (LER-affected 22 nm gate length GAA NW FETs).”

11) p. 8: Does the times required to train the MLP model include the time to produce the data for the training? This should be clarified.

Reply: The times required to train the MLP model do not include the time to generate the training data. That information is now included in the revised text (page 10) as follows:

“Note that times to generate the training data are not included in these experiments. ”

12) p. 11, line 268: A number is missing in (see Section ).

Reply: The text has been modified as follows: “In other words, computing the figures of merit for the 10 nm devices when using transfer learning is about 57,800x and 1,000x faster than MC and DD simulations, respectively (see Methodology section). ”

13) p. 11: How exactly is the Transfer Learning performed? More details how the TL when predicting the variability for the 10 nm gate NW FET is done when using the machine learning outcome for the 22 nm gate NW FET. For example, are hidden layers modified in some way, etc.?

Reply: When training for the 10 nm devices, we will retain the values of the model's trainable parameters from the previous model (22 nm devices) and use those initially instead of starting a training process from scratch. Therefore, the hidden layers can be modified, but the convergence is clearly faster. This information was included in the revised text (page 12), as follows:

“Next, we will apply a transfer learning approach to predict the figures of merit of the 10 nm devices using as starting point the trained models used for the 22 nm devices. It means that we will retain the values of the model’s trainable parameters from the previous model (22 nm devices) and use those initially instead of starting a training process from scratch.” 

14) A few comments could be added into Conclusions about applicability of the developed MLP network for the other sources of variability due to metal grains in the gate, line edge of the gate, and random dopands.

Reply: The following sentence has been added to Conclusions to highlight the applicability of the developed MLP network to othe sources of variability: 

“Finally, it is worth mentioning that the MLP architecture could also be applied (with an adequate calibration of the network hyperparameters and weights) to other relevant sources of variability affecting semiconductor devices, such as metal grain granularity, gate-edge roughness or random discrete dopants.”

---

## [Decision Letter · Decision Letter 1]

10 Jul 2023

A machine learning approach to model the impact of line edge roughness on gate-all-around nanowire FETs while reducing the carbon footprint

PONE-D-23-15843R1

Dear Dr. Loureiro,

We’re pleased to inform you that your manuscript has been judged scientifically suitable for publication and will be formally accepted for publication once it meets all outstanding technical requirements.

Kind regards,

Talib Al-Ameri, Ph.D

Academic Editor

PLOS ONE

Reviewers' comments:

Reviewer's Responses to Questions

**Comments to the Author**

1. If the authors have adequately addressed your comments raised in a previous round of review and you feel that this manuscript is now acceptable for publication, you may indicate that here to bypass the “Comments to the Author” section, enter your conflict of interest statement in the “Confidential to Editor” section, and submit your "Accept" recommendation.

Reviewer #1: All comments have been addressed

2. Is the manuscript technically sound, and do the data support the conclusions?

Reviewer #1: Yes

3. Has the statistical analysis been performed appropriately and rigorously? 

Reviewer #1: Yes

4. Have the authors made all data underlying the findings in their manuscript fully available?

Reviewer #1: Yes

5. Is the manuscript presented in an intelligible fashion and written in standard English?

Reviewer #1: Yes

6. Review Comments to the Author

Reviewer #1: I am very very very very happy with the revised version of the manuscript and recommend it publication.

7. PLOS authors have the option to publish the peer review history of their article (what does this mean?). If published, this will include your full peer review and any attached files.

Reviewer #1: **Yes: **Karol Kalna

---

## [Editor Report · Acceptance letter]

13 Jul 2023

PONE-D-23-15843R1 

A machine learning approach to model the impact of line edge roughness on gate-all-around nanowire FETs while reducing the carbon footprint 

Dear Dr. García-Loureiro:

I'm pleased to inform you that your manuscript has been deemed suitable for publication in PLOS ONE. Congratulations! Your manuscript is now with our production department. 

Kind regards, 

on behalf of

Dr. Talib Al-Ameri 

Academic Editor

PLOS ONE